# Deletion of Transglutaminase 2 from Mouse Astrocytes Significantly Improves Their Ability to Promote Neurite Outgrowth on an Inhibitory Matrix

**DOI:** 10.3390/ijms24076058

**Published:** 2023-03-23

**Authors:** Jacen Emerson, Thomas Delgado, Peter Girardi, Gail V. W. Johnson

**Affiliations:** Department of Anesthesiology and Perioperative Medicine, University of Rochester, 601 Elmwood Ave., Box 604, Rochester, NY 14620, USA

**Keywords:** transglutaminase 2, astrocytes, neurons, neurite outgrowth, CNS injury, Zbtb7a, transcriptional regulation

## Abstract

Astrocytes are the primary support cells of the central nervous system (CNS) that help maintain the energetic requirements and homeostatic environment of neurons. CNS injury causes astrocytes to take on reactive phenotypes with an altered overall function that can range from supportive to harmful for recovering neurons. The characterization of reactive astrocyte populations is a rapidly developing field, and the underlying factors and signaling pathways governing which type of reactive phenotype that astrocytes take on are poorly understood. Our previous studies suggest that transglutaminase 2 (TG2) has an important role in determining the astrocytic response to injury. Selectively deleting TG2 from astrocytes improves functional outcomes after CNS injury and causes widespread changes in gene regulation, which is associated with its nuclear localization. To begin to understand how TG2 impacts astrocytic function, we used a neuron-astrocyte co-culture paradigm to compare the effects of TG2−/− and wild-type (WT) mouse astrocytes on neurite outgrowth and synapse formation. Neurons were grown on a control substrate or an injury-simulating matrix comprised of inhibitory chondroitin sulfate proteoglycans (CSPGs). Compared to WT astrocytes, TG2−/− astrocytes supported neurite outgrowth to a significantly greater extent only on the CSPG matrix, while synapse formation assays showed mixed results depending on the pre- and post-synaptic markers analyzed. We hypothesize that TG2 regulates the supportive functions of astrocytes in injury conditions by modulating gene expression through interactions with transcription factors and transcription complexes. Based on the results of a previous yeast two-hybrid screen for TG2 interactors, we further investigated the interaction of TG2 with Zbtb7a, a ubiquitously expressed transcription factor. Co-immunoprecipitation and colocalization analyses confirmed the interaction of TG2 and Zbtb7a in the nucleus of astrocytes. Overexpression or knockdown of Zbtb7a levels in WT and TG2−/− astrocytes revealed that Zbtb7a robustly influenced astrocytic morphology and the ability of astrocytes to support neuronal outgrowth, which was significantly modulated by the presence of TG2. These findings support our hypothesis that astrocytic TG2 acts as a transcriptional regulator to influence astrocytic function, with greater influence under injury conditions that increase its expression, and Zbtb7a likely contributes to the overall effects observed with astrocytic TG2 deletion.

## 1. Introduction

Astrocytes play an indispensable role in maintaining a healthy environment for neuronal function in the central nervous system (CNS). They have primary roles in supporting synaptic structure and activity, as well as providing essential metabolic support to neurons [1,2,3]. In response to CNS injury, astrocytes take on a range of reactive phenotypes that influence neuronal survival and axonal regeneration. This response is a defensive reaction aimed at limiting tissue damage [4,5,6,7]. However, astrogliosis can also inhibit functional recovery [8,9,10,11,12]. Thus, reactive astrocytes have both beneficial and detrimental effects on the recovery process following injury. Although it is evident that astrocytes can exhibit either a more supportive or more harmful phenotype following an injury, the specific, intrinsic determinants that direct astrocytes towards either phenotype have not been well defined. 

Our previous studies have demonstrated that transglutaminase 2 (TG2) is a key variable in determining the molecular response of astrocytes to injury [13,14,15,16]. TG2 is a multifunctional protein; it catalyzes a calcium-dependent transamidation reaction, binds and hydrolyzes GTP and can function as a scaffold or linker protein [17,18,19,20,21]. Further, in astrocytes, TG2 is responsive to cell stressors, and expression levels are increased across CNS injury models in response to ischemia and inflammatory signals [22,23,24]. TG2 is primarily a cytosolic protein, but it can also be externalized and deposited into the extracellular matrix (ECM) [25] and localized to the nucleus where it is found in the chromatin fraction [26]. Given the fact that TG2 is found in the nucleus, it is not surprising that its ability to regulate gene transcription is well-documented across many cell types; however, these data are lacking in astrocytes, and specific mechanisms of transcriptional regulation have not been fully elucidated [21,27,28,29].

In vitro injury models show that TG2 negatively influences the response of astrocytes to an insult. Depletion or deletion of TG2 from astrocytes significantly increases their resistance to ischemic stress and their ability to protect neurons from ischemia-induced cell death [13,16,30]. Further, in an in vivo mouse model of spinal cord injury (SCI), selective deletion of TG2 from astrocytes (TG2fl/fl-GFAP-Cre+/−) significantly improved functional recovery [14]. GFAP, NG2 (chondroitin sulfate proteoglycan 4 [CSPG4]) and Sox9 immunoreactivity were also significantly decreased at the lesion site in the TG2fl/fl-GFAP-Cre+/− mice. These findings suggest that the deletion of TG2 from astrocytes increases their ability to promote neuronal recovery; however, this has not been directly demonstrated. Therefore, in this study, we used an in vitro astrocyte-neuron co-culture model to determine how the deletion of TG2 from astrocytes impacted their ability to support neurite outgrowth and synapse formation on permissive and inhibitory matrices. In addition, we provide evidence that TG2 may influence the response of astrocytes to injury in part by interacting with Zbtb7a, a ubiquitously expressed member of the POK (poxvirus and zinc finger and Kruppel)/ZBTB (zinc finger and broad complex, tramtrack, and bric a brac) family, which plays a key role in regulating gene expression [31]. The results of these studies clearly demonstrate that astrocytic TG2 plays a significant role in determining their ability to support the outgrowth of neurites on an inhibitory matrix. These data also indicate that TG2 may cooperate with Zbtb7a to determine the extent to which astrocytes can promote neurite outgrowth. Overall the results of these studies provide significant insights into the role of TG2 in determining the response of astrocytes to injury and the subsequent recovery process. 

## 2. Results

### 2.1. Neurite Outgrowth

In our previous study, we found that astrocyte-specific deletion of TG2 in a mouse model significantly improved the rate and extent of functional recovery following an SCI [14]; therefore, we hypothesized that TG2−/− astrocytes would promote neurite outgrowth and/or synapse formation to a greater extent compared to wild type (WT) astrocytes. Neurons were grown on either a permissive matrix (poly-D-lysine [PDL]) or CSPGs to reflect the growth-inhibitory extracellular environment of the SCI core and surrounding regions of reactive tissue [32,33]. In the absence of astrocytes, neurite outgrowth was significantly reduced for neurons grown on the CSPG inhibitory matrix compared to those grown on a permissive PDL matrix (Figure 1a,b). WT or TG2−/− astrocytes were then paired with neurons grown on the inhibitory matrix using the transwell paradigm. This co-culture method allows for the free exchange of soluble secreted factors and metabolites between neurons and astrocytes. The quantification of maximum neurite length showed that TG2−/− astrocytes promoted significantly greater neurite outgrowth on the inhibitory matrix compared to WT astrocytes (Figure 1a,b). Interestingly, WT and TG2−/− astrocytes promoted similar neurite outgrowth when the neurons were on a permissive matrix (Figure 5b). Additionally, we quantified the total number of primary neurites; neurons grown on the inhibitory matrix had significantly fewer primary neurites than those grown on the permissive matrix. This CSPG-induced reduction in primary neurite number was unaffected by the presence of astrocytes, either WT or TG2−/− (Figure 1a,c). Together, these results indicate that, compared to WT astrocytes, TG2−/− astrocytes are better able to facilitate neurite outgrowth of neurons on an inhibitory, injury-relevant, matrix, entirely through the exchange of soluble factors. It should be noted that in other experiments we found that the effect of CSPGs on neurite outgrowth was dependent on the seeding density of the neuron culture; at higher seeding densities on CSPGs, the effect of TG2−/− astrocytes on neurite outgrowth was no longer significantly different from WT astrocytes (Appendix A). 

### 2.2. Synapse Formation

To assess whether in addition to promoting greater neurite outgrowth, TG2−/− astrocytes also promoted greater functional connectivity, we measured the extent of synapse formation in our neuron-astrocyte co-cultures. For these experiments, we paired a neuron-seeded glass coverslip with an astrocyte-seeded glass coverslip, separated by paraffin pedestals [34], allowing for the analysis of astrocyte cell densities in addition to synaptic densities on the neuron coverslips. For our synapse assays, we used the same groups as in the neurite outgrowth experiments, except that neurons were seeded at higher densities. Given that it has been reported that E18 primary cortical neuron cultures undergo rapid development of synapses starting from DIV 7, with peaking network activity at DIV14 [35,36], we paired neurons with astrocytes from DIV 7 to DIV 12 to compare the contributions of each group of astrocytes to synapse development during this critical period. Consistent with the important role of astrocytes in synaptic development, we found that pairing neurons grown on a permissive matrix with astrocytes significantly increased the number of excitatory pre-synaptic marker, synaptophysin and post-synaptic marker, PSD-95 colocalizations (S/P). However, there was no difference in effect between the TG2−/− and WT astrocytes (Figure 2c). This analysis was run in parallel with a second set of excitatory pre- and post-synaptic markers (bassoon and homer, B/H), which interestingly showed a significant difference in colocalizations between WT and TG2−/− astrocytes (Figure 2d). After normalizing B/H colocalizations to the total number of bassoon puncta, the two groups were no longer significantly different, indicating that greater incoming pre-synaptic projections, marked by bassoon, may account for the significant increase in synapse formation among neurons paired with TG2−/− astrocytes. 

Neurons on the inhibitory matrix showed reduced colocalizations of S/P markers, by approximately half, compared with neurons on the permissive matrix. Interestingly, neurons on the inhibitory matrix displayed a more granular appearance of S/P markers rather than the clear punctate appearance seen for neurons on a permissive matrix. Additionally, neurites on CSPGs show a disorganized appearance with many self-synapses apparent (Figure 2a,b). The addition of astrocytes, WT or TG2−/−, had no significant effect on S/P synapse formation for neurons on an inhibitory matrix. Together, these results reinforce the well-understood concept that the presence of astrocytes facilitates synapse development [2]; however, this facilitation by astrocytes disappears for neurons grown on inhibitory CSPGs, regardless of the presence of astrocytic TG2. Although not quantified, CSPGs also appear to induce a near elimination of the homer puncta signal, of both large and small diameter, while the bassoon signal remains punctate and ubiquitous (Figure 2d).

Previously, we have reported significant differences in gene expression based on the presence or absence of TG2 in astrocytes [16]. RNAseq analyses of control and GFAP-Cre-TG2fl/fl spinal cords showed that changes in gene expression were only present in injury conditions, and interestingly all differentially expressed genes in the injured cords were upregulated in the mice with TG2 knocked out in astrocytes; the majority of these genes were associated with lipid metabolism [14]. RNAseq analyses of WT and TG2−/− astrocytes in culture showed a larger profile of up- and down-regulated genes in TG2−/− astrocytes, including those associated with the extracellular matrix, lipid metabolism, and cytoskeletal organization, with the majority of genes being upregulated [16]. Given the functional significance of gene regulation by TG2, we next wanted to investigate the mechanisms by which TG2 alters gene expression. Considering data from a previous yeast two-hybrid screen with TG2 as the bait [37] (Appendix A), we focused on the interaction of TG2 with Zbtb7a, a key regulator of gene expression. 

### 2.3. Immunoprecipitation and Colocalization

To investigate the potential contribution of TG2-Zbtb7a interactions to the outcomes we observed when TG2 expression was manipulated, we first carried out studies to confirm the interaction of TG2 with Zbtb7a. Fractionation studies of WT astrocytes demonstrated that both TG2 and Zbtb7a are present in the nucleus (Figure 3a). Considering previous work in the lab that localized TG2 in the chromatin fraction in nuclear extracts [26], colocalization of TG2 and Zbtb7a in the nucleus would support the hypothesis that they interact to modulate transcription [26]. Immunostaining of WT astrocytes for TG2 and Zbtb7a showed colocalization of the two proteins in the nucleus (Figure 3b), with only a small fraction of cellular TG2 in the nucleus. However, of the relatively small fraction of TG2 in the nucleus, approximately half colocalized with Zbtb7a. 

In contrast to TG2, the majority of Zbtb7a localized to the nucleus [38], with 5% colocalizing with TG2 (Figure 3b). To provide additional evidence of the TG2-Zbtb7a interaction, we carried out co-immunoprecipitation (co-IP) studies. V5-tagged TG2 and Myc/FLAG-tagged Zbtb7a were co-transfected in HEK293TN cells. Immunoprecipitation of V5-TG2 pulled down Myc/FLAG-Zbtb7a (Figure 3c), and vice versa (Appendix A), strongly supported an interaction between TG2 and Zbtb7a. Further, co-IP studies using nuclear fractions from WT astrocytes demonstrated that immunoprecipitated endogenous TG2 pulled down endogenous Zbtb7a (Figure 3d). Immunoprecipitation of endogenous Zbtb7a also resulted in TG2 co-precipitating (Appendix A). Taken together these data clearly indicate that TG2 and Zbtb7a interact.

### 2.4. Effect of Zbtb7a Manipulation on TG2−/− and WT Astrocytes

Having demonstrated a TG2-Zbtb7a interaction in the nucleus of astrocytes, we next explored the functional relevance of this interaction by analyzing the impact of Zbtb7a manipulation on astrocytes with or without TG2 present. For these studies, both WT and TG2−/− astrocytes were transduced with a shZbtb7a construct, a Zbtb7a overexpression construct, or a control vector. During the ten-day incubation period after transduction, we found that astrocytes across groups developed unique morphologies; therefore, we fixed and probed astrocytes for GFAP to evaluate alterations in the intermediate filament network (Figure 4b). Quantification of the GFAP network areas suggested the morphology of WT astrocytes was impacted to a significantly greater extent by Zbtb7a manipulation than the morphology of TG2−/− astrocytes (Figure 4c). Importantly, in control conditions, there was no difference between the GFAP network area of WT and TG2−/− astrocytes. In WT astrocytes alone, Zbtb7a knockdown significantly decreased the average GFAP network area, while Zbtb7a overexpression significantly increased the average GFAP network area. Although the GFAP network area is not a functional measurement, the differences observed indicate a differential impact of Zbtb7a manipulation on astrocyte cytoskeletal organization based on the presence or absence of TG2. 

To begin to investigate the functional relevance of the TG2-Zbtb7a interaction we manipulated Zbtb7a in WT and TG2−/− astrocytes followed by determining how this impacted their ability to support neurite outgrowth using the transwell paradigm. Similar to the morphology studies, Zbtb7a manipulation impacted the ability of WT astrocytes to support neurite outgrowth to a significantly greater extent than it impacted that of TG2−/− astrocytes (Figure 5). As mentioned above, on a permissive matrix there was no difference between the ability of WT and TG2−/− astrocytes to support neurite outgrowth. However, Zbtb7a knockdown trended toward a reduced ability of only WT astrocytes to support neurite outgrowth while Zbtb7a overexpression significantly and robustly increased the ability of WT astrocytes to support neurite outgrowth. Zbtb7a overexpression in TG2−/− astrocytes also seemed to increase their ability to promote neurite outgrowth, but not as robustly as was observed with WT astrocytes. 

## 3. Discussion

In a previous study, we demonstrated that astrocyte-specific deletion of TG2 resulted in a remarkably faster and overall greater functional recovery from SCI compared to WT mice [14]. This improved recovery was associated with reduced astrocytic reactivity in the injured spinal cord, as significantly less GFAP, NG2, and SOX9 immunoreactivity was evident at the injury site. These findings indicate that functional neuronal connections are reforming more rapidly and to a greater degree in the absence of astrocytic TG2. However, how TG2 deletion in astrocytes affects their ability to promote neurite outgrowth and/or synapse formation in injury conditions has not been previously explored. To analyze the mechanisms underlying improved functional recovery from SCI in mice with astrocyte-specific TG2 deletion [14], we used a neuron-astrocyte co-culture model that allows for the free exchange of soluble factors (trophic factors, metabolites, etc.) without the direct interaction between the two cell types [16]. 

Following SCI, astrocytes take on unique reactive states depending on their distance from the lesion, and this transformation is associated with increased astrocytic secretion of ECM components, including CSPGs [8,33,39]. CSPGs are common components of the healthy adult neural ECM, which can be growth-supportive or growth-inhibitory for regenerating axons depending on the specific member of the CSPG family and sulfation patterns [40,41]. After SCI, inhibitory CSPGs are densely deposited in the ECM of the lesion core and penumbra and they inhibit neuronal regeneration across the lesion [42]. To simulate the growth-inhibitory ECM of SCI, we grew neurons on an inhibitory CSPG matrix that is a well-accepted model of an injury-induced inhibitory ECM [43,44,45]. Considering that neurite outgrowth and synapse formation are necessary for re-establishing a neuronal network after injury, we compared the ability of WT and TG2−/− astrocytes to support these two processes on an inhibitory matrix.

On the inhibitory, but not the permissive matrix, TG2−/− astrocytes supported neurite outgrowth to a significantly greater extent than WT astrocytes. Considering the transwell design of this assay, this shows that TG2−/− astrocytes are able to better support neurons in overcoming growth-inhibitory signals entirely through the free exchange of soluble factors within the media. This raises the question of what cellular processes, downstream of CSPG signaling, are being overcome by exchanging soluble factors with astrocytes. Signaling pathways downstream of CSPG-specific receptors, LAR and PTPRσ, have yet to be fully characterized, but the activation of these receptors is associated with the inhibition of PI3K/AKT signals and activation of RhoA/ROCK signaling, which both contribute to growth inhibition [46]. Additionally, the activation of PTPRσ by inhibitory CSPGs around the growth cone leads to decreased autophagic flux and the formation of dystrophic end bulbs [40,47]. It is currently unclear to what degree neurons on inhibitory matrices experience unique stressors or energetic and resource demands, as may be speculated by CSPG’s effects on autophagy. Indeed, TG2−/− astrocytes may better maintain these unique neuronal resource requirements compared to WT astrocytes. However, our observations that high-density neuron cultures overcome CSPG growth inhibition (Appendix A), while still having largely disorganized neuritic structures, may indicate that local growth factor production among dense groups of neurons is alone sufficient to overcome growth inhibition in our in vitro model, with no observable improvements in what may be disruptions in neurite cytoskeletal organization or tropism of neurite growth (as replicated in our synapse studies). 

As expected, astrocytes promoted synapse formation on a permissive matrix, however, there was no difference between WT and TG2−/− astrocytes in their ability to increase S/P synapse colocalization. Yet, on a permissive matrix, there was a significant difference between the two astrocyte groups with B/H synapse formation. Further analysis of the data revealed that this effect may be accounted for by a greater number of bassoon-containing pre-synaptic projections to TG2−/− astrocyte-paired neurons. As mentioned above, TG2 deletion in astrocytes leads to a differential regulation in lipid metabolism, and previous data have shown that astrocytic lipid metabolism is critical for pre-synaptic function and development [48]. Therefore, our B/H synapse data may be partially explained by enhanced lipid/cholesterol supply to growing neurites by TG2−/− astrocytes. We need to further replicate these data to confirm a difference in effect between marker sets. Interestingly, on an inhibitory matrix, astrocytes, both WT or TG2−/−, did not promote greater synapse formation compared to no-astrocyte controls. This suggests that the improved recovery observed in the astrocyte-specific TG2 deletion mice after SCI was not due to astrocyte-secretory mechanisms that directly affect synapse formation. While TG2−/− astrocytes alone cannot improve synapse formation on an inhibitory matrix through the free exchange of soluble factors, perhaps direct contact and pericellular signaling between these astrocytes and neurons can expedite synapse formation and injury recovery. Additionally, in SCI, if there truly is no differential effect on synapse formation in an inhibitory environment, it is likely that the ability of TG2−/− astrocytes to support axonal regeneration across dense areas of the inhibitory matrix would allow for synapse formation in permissive matrices away from the injury site and improve overall functional recovery. CSPGs also induced apparent dysfunction in both neuritic structures (also seen in our neurite outgrowth studies) and synaptic protein aggregation, which was unaffected by the presence of astrocytes. These effects may be produced by signaling pathways downstream of CSPG receptors, and this emphasizes the importance of the further characterization of these pathways. 

RNAseq analyses of TG2−/− and WT astrocyte cultures, and of injured spinal cords from WT mice and mice with astrocyte-specific TG2 deletion, strongly suggest that TG2 acts predominantly, but not exclusively, to repress gene expression [14,16]. Interestingly, this difference in gene expression was not observed in uninjured spinal cords from astrocyte-specific TG2 deletion mice and WT mice [14]. Inflammatory signals, which occur subsequent to CNS injury, can cause astrocytes to take on reactive phenotypes [10]. These injury signals also directly increase TG2 expression in astrocytes, which may then influence the development of reactive astrocyte phenotypes and functions [49,50]. Given the results of previous studies, we speculated that the ability of astrocytic TG2 to mediate injury responses was in part due to its ability to direct gene expression by interacting with transcriptional regulators, and based on an earlier yeast two-hybrid study, we identified Zbtb7a as a possible factor that is modulated by TG2. 

Zbtb7a is a transcription factor with a DNA binding domain that has been shown to modulate the expression of genes regulated by SP1, E2F-4 and NF-κB, binding motifs also common to many of the genes downregulated in the presence of TG2 in our RNAseq data sets [14,16,31,51,52]. Zbtb7a can enhance gene expression by assisting in the relaxation of chromatin [31]. Additionally, Zbtb7a can interact with the Sin3a repressor complex to attenuate gene expression [51]. We have found that TG2 also likely binds SAP18 (Appendix A)—a component of the Sin3a complex [53]. Using immunocytochemical fluorescent colocalization and protein immunoprecipitation, we were able to confirm that TG2 and Zbtb7a interact in the nucleus of astrocytes. Interestingly, approximately half of the small amount of TG2 that enters the nucleus interacts with Zbtb7a, suggesting that Zbtb7a is a key component of TG2 transcriptional complexes.

To assess the functional implications of the interaction between TG2 and Zbtb7a, we modulated the expression of Zbtb7a in TG2−/− and WT astrocytes and examined outcomes. Interestingly and unexpectedly, Zbtb7a manipulation differentially impacted the morphology of astrocytes based on the presence of TG2. These data suggest that Zbtb7a significantly influences the astrocytic cytoskeleton; an effect of Zbtb7a that has not been previously reported. In addition, although the GFAP network was robustly influenced by the knockdown or overexpression of Zbtb7a in WT astrocytes, only very modest changes were observed in the absence of TG2. This would seem to indicate that Zbtb7a plays the primary role in mediating these changes in the cytoskeleton while TG2 is a modulator. Nonetheless, this finding demonstrates that TG2 cooperates with Zbtb7a to mediate outcomes in astrocytes. To better understand this phenomenon, we looked back at the major gene groups impacted by TG2 in previous RNAseq experiments. Two major groups of genes regulated by TG2: lipid metabolism and cytoskeletal-related genes are possible factors in determining astrocytic morphology [14]. While astrocytic morphology is not a direct measure of their ability to support neuronal health, these data are important to our studies first in showing that Zbtb7a has a differential impact on astrocytes based on the presence of TG2, and it may indirectly reflect a change in astrocytic metabolism or function which is important in their interaction with neurons.

Pairing these astrocytes with neurons also revealed a differential impact of Zbtb7a manipulation based on the presence of TG2. As with the GFAP cytoskeletal effects, knocking down or overexpressing Zbtbt7a had a much greater effect on the ability of the WT astrocytes to support neurite outgrowth compared to TG2−/− astrocytes. This again suggests that TG2 mediates the effect of Zbtb7a, perhaps by preventing Zbtb7a from affecting the transcription of genes that are not necessarily directly regulated by TG2. It can be speculated that the interaction of TG2 with Zbtb7a may prevent it from interacting with chromatin areas where it usually binds to facilitate gene transcription [31]. It is possible that the differential effects of Zbtb7a manipulation, with and without TG2 present, on the astrocytic cytoskeleton and the ability of astrocytes to promote neurite outgrowth are related to the TG2 differentially regulated pathways we previously identified [14,16], but further studies are necessary to explore this supposition. 

Considering these findings and previous studies, we hypothesize that TG2 represses genes that enable astrocytes to support neuronal health, which is exacerbated in injury conditions where TG2 expression is increased [50]. Experiments manipulating Zbtb7a levels in TG2−/− and WT astrocytes suggest TG2 antagonizes the effects of Zbtb7a on the cytoskeleton and the ability of astrocytes to support neurite outgrowth. It may be speculated that Zbtb7a enhances the expression of genes involved in the astrocytic response to injury, perhaps by relaxing chromatin [31] and that this process is attenuated by the presence of TG2. Further studies will focus on understanding the mechanisms by which TG2 and Zbtb7a interact to mediate the observed differences in gene expression.

## 4. Materials and Methods

### 4.1. Animals

All mice and rats were maintained on a 12-h light/dark cycle with food and water available ad libitum. The procedures with animals were in accordance with guidelines established by the University of Rochester Committee on Animal Resources. The studies were carried out with approval from the Institutional Animal Care and Use Committee. WT C57BL/6 mice were originally purchased from Charles River Laboratories. TG2−/− mice on a C57Bl/6 background were described previously [13]. Timed pregnant Sprague Dawley rats were obtained from Charles River Laboratories.

### 4.2. Cell Culture

Primary cortical neurons were prepared from Sprague Dawley rat embryos at embryonic day 18 (E18) and cultured as previously described with some modifications [54]. To prepare the coverslips/wells, PDL (Sigma, St. Louis, MO, USA P6407) was diluted in PBS to a concentration of 20 µg/mL and added to the wells for 4 h. The wells were either rinsed and stored with PBS, or after rinsing, CSPGs (Millipore, St. Louis, MO, USA CC117) in PBS (2.5 µg/mL) were added and incubated overnight to coat the coverslips. All wells and coverslips were rinsed with PBS prior to plating the neurons. To prepare the neurons, a pregnant rat was euthanized using CO_2_, followed by rapid decapitation. Embryonic brains were isolated, cerebral cortices dissected, and meninges were removed. Cerebral cortices were then digested in trypsin-EDTA (0.05%) (Corning, Corning, NY, USA 25-053-Cl) for 15–20 min in a 37 °C water bath. Following gentle trituration, neurons were plated in Neuron Plating media consisting of MEM (Gibco, Waltham, MA, USA 42360032) supplemented with 5% FBS, 20 mM glucose, and 0.2% Primocin (InvivoGen, San Diego, CA, USA ant-pm-2) at a density of 12,000 cells/cm^2^ on the coated coverslips for neurite outgrowth and for synaptic analyses 24,000 cells/cm^2^. Four to five hours later, the media was replaced with Neurobasal media (Gibco, 21103-049) containing 2% B27 (Gibco, 17504-044), 0.5 mM Glutamax (Gibco, 35050-061) and 0.2% Primocin (Neuron Growth media). Neurons were incubated at 37 °C/5% CO_2_ and experiments began at DIV 1.

Primary astrocytes were cultured at post-natal day 0 from either wild-type C57BL/6 or TG2−/− mouse pups as previously described [13]. In brief, the brains were dissected, meninges removed, and cortical hemispheres were collected. Following trituration of the cells, they were plated onto culture dishes in MEM supplemented with 10% FBS, 33 mM glucose, 1 mM sodium pyruvate (Gibco, 11360-070), and 0.2% Primocin (Glial MEM). Twenty-four hours later, the dishes were shaken vigorously and rinsed to remove debris and other cell types. Astrocytes were maintained at 37 °C/5% CO_2_ for 7–8 days, frozen in Glial MEM containing 10% DMSO, and stored in liquid nitrogen. To seed the astrocytes onto the transwells, wild-type or TG2−/− astrocytes were thawed and plated onto 6 cm dishes. Once at 80–90% confluency, cells were split and seeded onto transwell inserts (6.5 mm) with a membrane pore size of 1.0 μm in Neuron Growth media 24 h prior to pairing with the neurons. On neuron DIV 1, the inserts were placed over the neurons on coverslips in a 12-well plate and the neurons received a half media change with astrocyte-conditioned media. The cell pairs were incubated for 96 h and coverslips with neurons were collected for analysis of neurite outgrowth. For analyses of synapse formation, astrocytes were paired with neurons on DIV 7 and collected on DIV 12. In this assay, astrocytes were seeded on PDL-coated glass coverslips and sandwiched with neuron-seeded coverslips, separated only by paraffin pedestals as described previously [34].

### 4.3. Neurite Outgrowth Analyses

Coverslips of neurons from the transwell co-cultures were washed three times with PBS, followed by fixation with 4% paraformaldehyde and 4% sucrose in PBS for 5 min. After three washes with PBS, the cells were permeabilized with 0.25% Triton X-100 in PBS and blocked with PBS containing 5% BSA and 0.3 M glycine. MAP2 primary antibody (1:200) (Cell Signaling, Danvers, MA, USA #8707S) was diluted in blocking buffer and incubated overnight on the coverslips. The next day, the coverslips were washed three times and incubated in Alexa Fluor 594 donkey anti-rabbit (Invitrogen, Waltham, MA, USA A21207) for 1 h. Coverslips were then counterstained with Hoechst 33342 (1:10,000) and mounted using Fluoro-gel in TES Buffer (Electron Microscopy Sciences, Hatfield, PA, USA 17985-30). The slides were imaged using a Zeiss Observer D1 microscope with a 40× objective.

Ten to fifteen neurons per coverslip were imaged for each condition. Images were processed by Image J Fiji using the Simple Neurite Tracer plugin. Neurites were traced using a scale of 6.7 pixels/µm. For the max neurite length studies, the longest neurite of each neuron was recorded. The number of neurite paths that directly extended from the soma was counted for each neuron to determine the number of primary neurites.

### 4.4. Synaptic Analyses

Coverslips of the neurons and astrocytes, from the sandwich method of coincubation, were processed at the same time. Coverslips were washed three times with PBS, followed by fixation with 2% paraformaldehyde and 4% sucrose in PBS for 5 min. After three washes with PBS, the cells were permeabilized with 0.1% Triton X-100 in PBS and blocked with PBS containing 5% BSA and 0.3 M glycine. For neurons, the primary antibodies Synaptophysin (1:200) (Sigma #S5768), PSD-95 (1:250) (Cell Signalling #3450), Bassoon (1:200) (Cell Signaling #6897), and Homer (1:200) (Santa Cruz, Dallas, TX, USA #17842) were diluted in blocking buffer and incubated overnight on the coverslips in the combinations indicated. The astrocytes were probed with GFAP (1:300) (Sigma #G3893) or Vimentin (1:100) (Cell Signaling #5741S) primary antibodies to track confluencies of the astrocyte coverslips throughout experimental conditions. The next day, the coverslips were washed three times and incubated in Alexa Fluor 594 and Alexa Fluor 488 secondary antibodies (ThermoFisher, Waltham, MA, USA) for 1 h. Coverslips were mounted using Fluoro-gel in TES Buffer. The slides were imaged using an Olympus Scanning Confocal Microscope (FV1000) with a 60× Oil Objective (1.35 NA), at a 10 μs/pixel scanning speed and Kalman averaging value of 4. For each neuron, z-stack images were taken at 0.5 μm step size to capture synapse puncta throughout all planes of the cell; 5–10 neurons per coverslip were imaged, and at least three coverslips were used for each group. In Imaris, images were deconvoluted and max intensity projections were created. Before counting synaptic puncta on neuronal processes, masks were created using the surface function to remove the cell soma from the images, leaving only the neuritic processes; quantification was limited to a 60 μm radius around the center of the cell soma. The spots function was used to identify synaptic puncta based on quality and mean fluorescent intensity limits. The settings for identification of spots by quality were kept consistent across groups at 155 for the post-synaptic markers and 100 for the pre-synaptic markers, while for intensity settings, spots were identified subjectively, and as consistently as possible, in the range of 300 to 750, due to differences in background fluorescence across replicates. The spots colocalization function (spot distance threshold of 0.5 μm) was used to quantify pre- and post-synaptic marker colocalization.

### 4.5. Constructs 

V5 tagged human TG2 in pcDNA and in the lentiviral vector FigB has been described previously [54,55]. The FLAG/Myc-tagged Zbtb7a construct was purchased from Origene (RC222759). The Zbtb7a shRNA (5′-GCCAGGAGAA GCACTTTAAG-3) was cloned into the pHUUG vector (a generous gift from Dr. C. Proschel). The PSP lentiviral packaging construct and VSVG lentiviral envelope construct were also generous gifts from Dr. C. Proschel. The PMDG 8.9 and PMDG VSVG lentiviral packaging and envelope constructs as well as the human Zbtb7a lentiviral construct were generous gifts from Dr. Jasper Yik [56].

### 4.6. Lentiviral Transduction

Lentiviruses were packaged in HEK293TN cells as described previously [13]. In brief, Zbtb7a shRNA or scrRNA constructs were co-transfected into HEK293TN cells with PSP and VSVG viral coat and packaging proteins. Zbtb7a overexpression lentivirus was made by co-transfecting the Zbtb7a lentiviral construct, PMDG 8.9 and PMDG VSVG into HEK293TN cells. HEK293TN cells were kept at 33 °C/5%CO_2_ for 72 h. After 72 h the virus-containing media was collected and filtered with a 0.22 µm filter. The viral particles were then pelleted by centrifugation at 35,000× *g* for 4 h at 4 °C. Viral pellets were collected in Neurobasal media and kept at −80 °C for later use. For viral transduction, the thawed virus was added to WT or TG2−/− astrocytes plated the day prior. Half media changes were conducted every 4 days and the cells were fixed for staining, split onto transwells, or fractionated for immunoblotting 10 days after transduction.

### 4.7. Nuclear Fractionation and Immunoblotting

To separate nuclear and cytoplasmic fractions from WT and TG2−/− astrocytes, the NE-PER nuclear and cytoplasmic extraction kit (Thermo Scientific 78833) was used per the manufacturer’s protocol. Protein concentrations were determined using a BCA assay. Samples were diluted to 1 μg/μL in 1X SDS sample buffer and incubated for 10 min at 100 °C. The samples were resolved on 12% SDS-PAGE gels and proteins were transferred to a nitrocellulose membrane. Membranes were blocked in 5% milk in Tris-Buffered Saline with Tween20 (TBS-T) (20 mM Tris base, 137 mM NaCl, 0.05% Tween20) for 1 h at room temperature. After blocking, primary antibodies against TG2 (rat anti-mouse TG2 antibody, TGMO1, [57]), Zbtb7a (Hamster monoclonal antibody 13E9, Santa Cruz Biotechnology sc-33683), or Beta Tubulin (rabbit polyclonal antibody, Proteintech, Rosemont, IL, USA 10094-1-AP) were added to the blots in blocking buffer and incubated at 4 °C overnight. The next day blots were washed with TBS-T and incubated for 1 h at room temperature with HRP-conjugated secondary antibody. The blots were washed with TBS-T before being visualized with an enhanced chemiluminescence reaction. 

### 4.8. Immunocytochemistry for Astrocytes

Astrocytes were plated on 18 mm coverslips and grown in glial MEM at 37 °C/5%CO_2_. Once confluent, astrocytes were washed in TBS. After washing the astrocytes were fixed in ice-cold methanol for 10 min at room temperature. The cells were again washed in TBS before being blocked and permeabilized in 3% BSA, and 0.05% Triton-X in TBS for 30 min at room temperature. After being blocked and permeabilized, the astrocytes were labeled with sheep anti-TG2 (R&D Systems, Minneapolis, MN, USA AF5418) and hamster anti-Zbtb7a or mouse anti-GFAP (Sigma G3893). The primary antibodies were added in 5% BSA and incubated overnight at 4 °C. Cells were then washed with TBS before being incubated with Alexa Fluor 594 conjugated rabbit anti-hamster antibody (Jackson ImmunoResearch, West Grove, PA, USA 307-585-003) and Alexa Fluor 488 conjugated rabbit anti-sheep antibody (Jackson ImmunoResearch 313-545-045) for colocalization experiments or Alexa Fluor 488 conjugated anti-mouse antibody (Invitrogen 21042) for GFAP immunostaining in 5% BSA for 1 h at room temperature. The cells were washed in TBS before being stained with DAPI diluted in TBS for 10 min at room temperature. The coverslips were then mounted on slides using Fluor-Gel with TES Buffer (Electron Microscopy Sciences 17985-30). 

### 4.9. GFAP Network Quantification

Astrocytes immunostained for GFAP were visualized using a Zeiss Observer D1 microscope. The cells were viewed with a 40× oil objective and images were captured using Zen 3.4 (Blue Edition) software. The acquisition protocol utilized the Alexa Fluor 488 channel, at 15% LED intensity and a 500 ms exposure, and the DAPI channel, at 15% LED intensity and a 150 ms exposure. After being captured, the area of the GFAP network was measured using the analysis software in Zen. The total network area was measured using auto analysis to map the entire area of the fluorophore signal, regardless of intensity, and provide a datapoint in square microns based on the scale of the image.

### 4.10. Colocalization Analyses 

Immunostained astrocytes were imaged using an Olympus Scanning Confocal Microscope (FV1000) with a 60× oil objective (1.35 NA), at a 10 μs/pixel scanning speed and Kalman averaging value of 2. For each astrocyte, z-stack images were taken at 0.5 μm step size to capture TG2 and Zbtb7a signal throughout all planes of the nucleus. Approximately five astrocytes per coverslip were imaged. In Imaris, images were deconvoluted and max intensity projections were created. Before the colocalization of Zbtb7a and TG2 signals were quantified, the entire image except for any nuclei was masked. Once only the nuclear signal remained the background subtraction feature was used in the TG2 channel to minimize non-specific signal. Puncta with a diameter of less than 3 μm were filtered out. The co-localization function in Imaris was used to determine the proportion of each signal overlapping with the other. Within the co-localization function intensity thresholds were set at 320 for TG2 and 260 for Zbtb7a. Co-localization values of each channel were reported as Mander’s coefficients.

### 4.11. Co-Immunoprecipitation

For exogenous immunoprecipitation (IP), HEK293TN were transfected with V5-TG2 and FLAG/Myc-Zbtb7a constructs using PolyJet transfection reagent (Signagen, Frederick, MD, USA #SL100688) following the manufacturer’s protocol. After 24 h the cells were lysed and collected in IP lysis buffer (150 mM NaCl, 50 mM Tris-HCl, 1 mM EDTA, 1 mM EGTA, 0.5% NP-40 in PBS). For endogenous immunoprecipitation, WT astrocytes were fractionated as described above and the nuclear fractions were collected. Protein concentrations of endogenous and exogenous samples were measured using a BCA assay. Five hundred micrograms of HEK293TN cell lysate (exogenous), 400 µg of astrocyte cytosolic protein, or 110 µg of nuclear protein (endogenous) were used for IP. To the exogenous protein samples, 8 µL of rabbit anti-V5 tag antibody (Cell Signaling 13202S) was added to each sample. To the endogenous protein samples, 4 µL of either mouse anti-TG2 antibody (Novus, Centennial, CO, USA NBP2-26458) or hamster anti-Zbtb7a antibody (Invitrogen 14-3309-82) was added to each sample. Once primary antibodies were added, the samples were incubated on a rotator at 4 °C overnight. IgG control samples were incubated with an equivalent amount of normal rabbit (Millipore 12-370) or mouse (Millipore 12-371) IgG antibodies. After 18 h, 30 µL of Pierce protein A/G magnetic agarose, (Thermo Scientific 78609) for the exogenous samples, or 30 µL of Pierce protein L magnetic agarose beads, (Thermo Scientific 88850) for the endogenous samples, washed in IP wash buffer (2 mM EDTA, 0.1% NP-40 in PBS) and blocked in 1% BSA in PBS, were added. After a 6 h incubation, rotating at 4 °C, the samples were thoroughly washed in IP wash buffer and then in IP lysis buffer. After washing, beads were incubated in 30 µL of 2.5× SDS in IP lysis buffer for 10 min at 100 °C. Samples were then immunoblotted as previously described.

### 4.12. Statistical Analysis

GraphPad Prism was used to report the raw data and perform statistical analysis. The mean values and standard error of the mean were calculated for each group. A two-way ANOVA was used to compare more than two groups with two independent variables and an unpaired *t*-test was used to compare two groups, and levels of significance were set at * *p* < 0.05, ** *p* < 0.01, *** *p* < 0.001, **** *p* < 0.0001. 

## Figures and Tables

**Figure 1 ijms-24-06058-f001:**
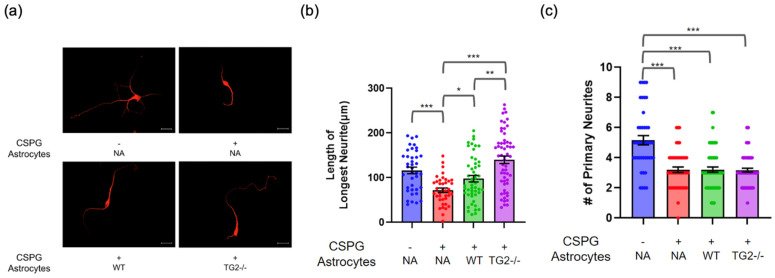
TG2−/− astrocytes promote neurite outgrowth to a greater extent than wild-type (WT) astrocytes on an inhibitory matrix (CSPG). (**a**) Representative MAP2 images of neurons grown on poly-D-lysine (PDL) only (−) or PDL+CSPG (+) matrix and paired with WT, TG2−/−, or no astrocytes (NA) (scale bar = 20 µm) (**b**) Quantitation of neurite length on CSPG (n = 39–57 neurons per condition from three independent biological replicates, two-way ANOVA * *p* < 0.05, ** *p* < 0.01, *** *p* < 0.001). (**c**) Quantitation of primary neurite number on CSPG (n = 43–68 neurons per condition from three independent biological replicates, two-way ANOVA *** *p* < 0.001).

**Figure 2 ijms-24-06058-f002:**
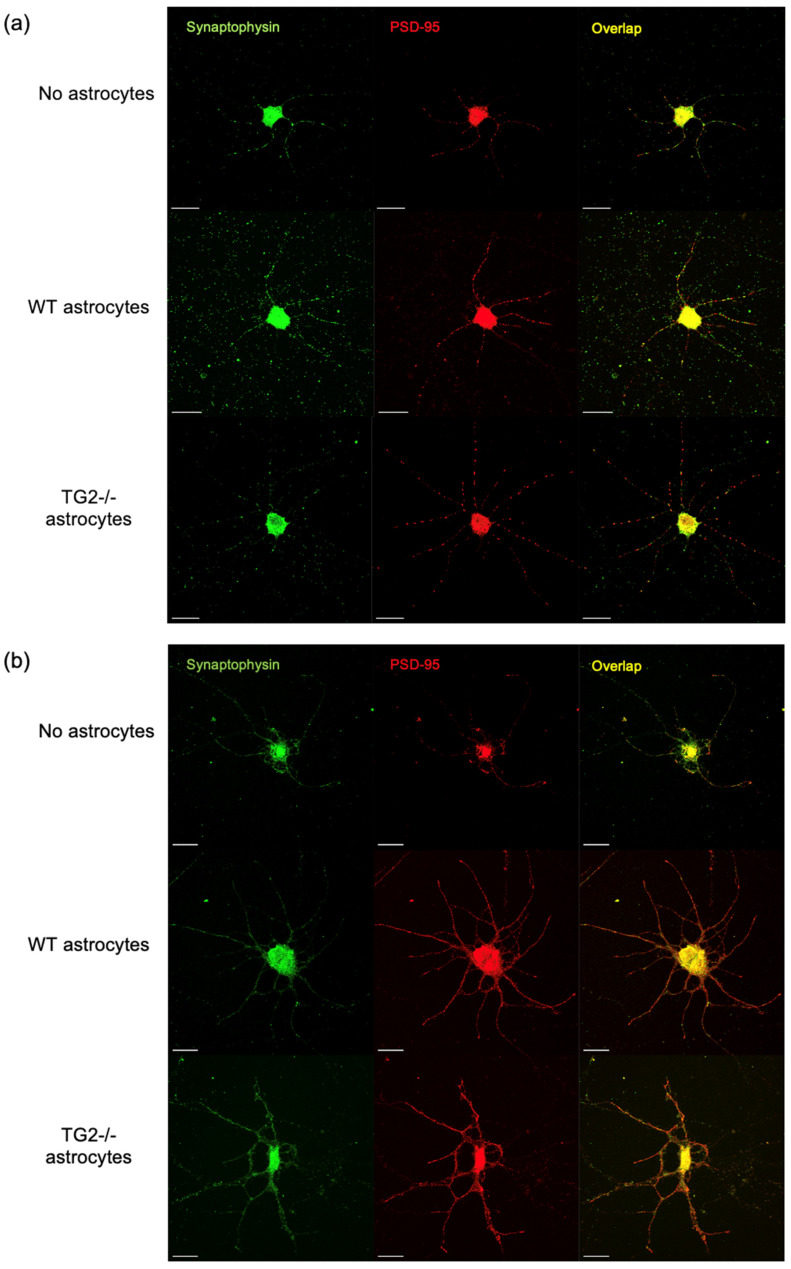
Astrocytes facilitate synapse formation among neurons on a permissive, but not inhibitory matrix, independent of the presence of TG2. (**a**) Images of PSD95/Synaptophysin immunostaining of neurons grown on poly-D-lysine (PDL) with no astrocytes (NA) or with WT or TG2−/− astrocytes (scale bar = 20 µm), (**b**) Images of PSD95/synaptophysin immunostaining of neurons grown on CSPG inhibitory matrix without astrocytes or with WT or TG2−/− astrocytes (scale bar = 20 µm), (**c**) Quantification of neuronal synapses approximated by PSD95 puncta collocated with Synaptophysin puncta exclusively in neurites; intensity-based identification of spots was used to more accurately capture puncta in CSPG conditions (18–24 neurons per condition from two independent biological replicates, Two-way ANOVA, **** *p* < 0.0001), (**d**) Representative images of Homer/Bassoon immunostaining for neurons grown on PDL and PDL+CSPG (scale bar = 20 µm), and quantification of synapses on PDL approximated by Homer puncta collocated with Bassoon puncta exclusively in neurites (22–23 neurons per condition from two independent biological replicates, unpaired *t*-test, *** *p* < 0.001).

**Figure 3 ijms-24-06058-f003:**
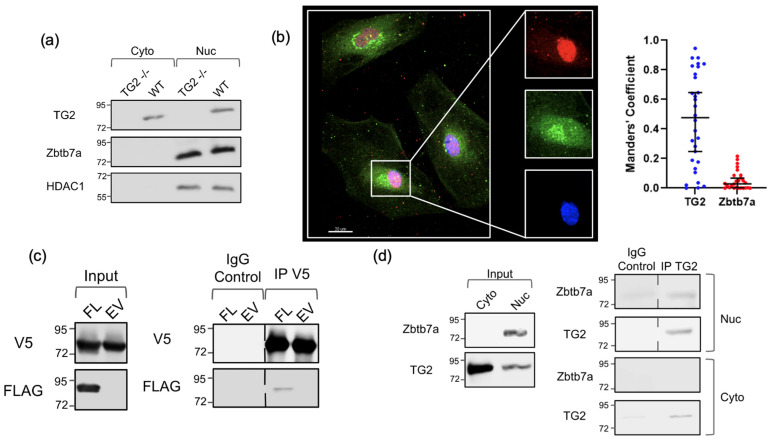
TG2 and Zbtb7a interact. (**a**) Nuclear (Nuc) and cytosolic (Cyto) fractions of wild type (WT) and TG2 knockout (TG2−/−) astrocytes. Intensity of bands are not representative of the proportion of the total protein in the cytosolic/nuclear fractions. (**b**) Immunocytochemistry showing co-localization of TG2 (green) and Zbtb7a (red) in nucleus (blue) of astrocytes. Quantification of co-localization of nuclear TG2 and Zbtb7a signal with the median Manders’ coefficient and 95% CI plotted (n = 30 from 2 independent biological replicates). (**c**) Left panel: Input controls of V5-TG2 and FLAG-Zbtb7a (FL) or empty vector (EV) transfected in HEK293TN cells. Right panel: Immunoprecipitation of V5-TG2 (V5) pulls down FLAG-Zbtb7a (FLAG). (**d**) Immunoprecipitation of endogenous TG2 from astrocyte nuclear fraction (nuc) pulls down Zbtb7a. Left panel: input, Right panel: immunoprecipitation with an IgG control or a TG2 antibody. In (**c**,**d**) vertical dashed lines indicate that intervening lanes were removed.

**Figure 4 ijms-24-06058-f004:**
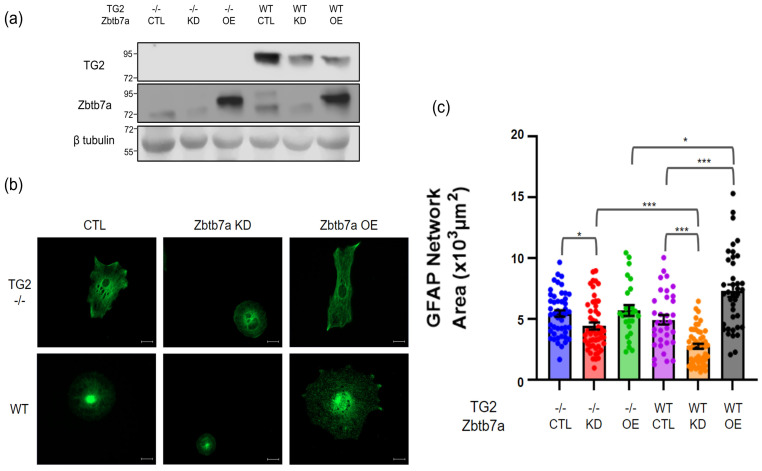
Manipulating the expression of Zbtb7a in WT and TG2−/− astrocytes differentially impacts morphology. (**a**) Immunoblots showing knockdown (KD) or overexpression (OE) of Zbtb7a, compared to control (CTL), in WT and TG2−/− astrocytes. Since the Zbtb7a OE virus encodes human Zbtb7a, the Zbtb7a bands in these samples migrated slightly faster than bands in the CTL and KD samples from mouse astrocytes. (**b**) Representative images of the astrocyte GFAP network in the different conditions (scale bar = 20 µm) and (**c**) Quantitation of GFAP network area (n = 34–50 astrocytes per condition from two independent biological replicates, Two-way ANOVA * *p* < 0.05, *** *p* < 0.001).

**Figure 5 ijms-24-06058-f005:**
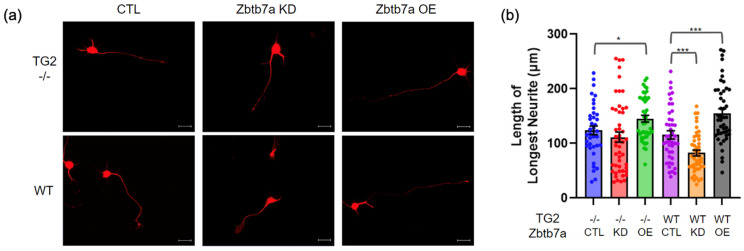
Knocking down or overexpressing Zbtb7a in WT or TG2−/− astrocytes differentially impacts their ability to promote neurite outgrowth on a permissive substrate (poly-D-lysine). (**a**) Representative MAP2 images of neurons that were paired with Zbtb7a control (CTL), knockdown (KD) or overexpression (OE) transduced TG2−/− or wild type (WT) astrocytes (scale bar = 20 µm). (**b**) Quantitation of neurite length on the permissive substrate (n = 38–49 neurons per condition from 2 independent biological replicates, Two-way ANOVA, * *p* < 0.05; *** *p* < 0.001).

## Data Availability

Data is contained within this article and Appendix A.

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
