# Peer review of "Deletion of Transglutaminase 2 from Mouse Astrocytes Significantly Improves Their Ability to Promote Neurite Outgrowth on an Inhibitory Matrix"

_ijms, 2023, doi:10.3390/ijms24076058_

Round 1

Reviewer 1 Report

In this manuscript, the authors present new pieces of evidence on how the presence of TG2 in astrocytes negatively influences neurons by affecting the neurite outgrowth upon injury in an inhibitory ECM. In addition, the interaction between ZbTb7a and TG2 was experimentally confirmed providing a potential explanation regarding the mechanism of the TG2 effect.

The manuscript is well-written, logical and contains only a few mistyping.

Major problems:

1. Probably due to the editing, one figure covers a part of the text and it is not possible to read and check (page 6, lane 172-195). Based on the figure legend, probably the pictures of Fig3 and 4 are mixed up.

2. What is the reason, that in Fig3d, where proteins were immunoprecipitated using an anti-TG2 antibody from the cytoplasm the TG2 is not detectable although based on the input, the cytoplasm contains a huge amount of TG2? Is there any other repetition to clarify this issue? Or is it possible to publish the experiments using other TG2-specific, high-affinity antibody? In the supplement to figure 3d TG2 band is also not clearly visible.

Minor comments, and questions which could contribute to the improvement of the manuscript:

1. Please, make it more clear in the abstract (and in the title), that the study is based on mouse cells/model.

2. The number of independent repetitions of experiments is not obvious. Please, label in the text, or method or figure legends where it is applicable.

3. The list of abbreviations or completion of figure legends with them would make easier the understanding.

4. Labeling the molecular weight of protein markers is necessary to judge the original images due to probably unspecific bands.

5. Some additional explanations and labels would help the complete understanding of Yeast Two-Hybrid results in the supplement.

6. Page 8 lane 231-232: Maybe it could be written: ”…, but not so significantly,…”. based on the results in Fig5b (* vs. ***).

7. It is not completely clear, why has TG2 a necessarily inhibitory effect on gene expression. Is it possible, that the reason for increased gene expression is compensation due to the loss of TG2?

Author Response

Reviewer #1

  1. Probably due to the editing, one figure covers a part of the text and it is not possible to read and check (page 6, lane 172-195). Based on the figure legend, probably the pictures of Fig3 and 4 are mixed up.

We apologize for this error. It was difficult to position and correctly size images, especially multi-paneled micrographs, using the journal’s template which resulted in this error in formatting.  We have now put more effort into correctly formatting the manuscript and have carefully checked it before resubmitting it.

  1. What is the reason, that in Fig3d, where proteins were immunoprecipitated using an anti-TG2 antibody from the cytoplasm the TG2 is not detectable although based on the input, the cytoplasm contains a huge amount of TG2? Is there any other repetition to clarify this issue? Or is it possible to publish the experiments using other TG2-specific, high-affinity antibody? In the supplement to figure 3d TG2 band is also not clearly visible.

When doing the immunoprecipitation shown in Fig3d 10 plates worth of fractionated lysates were combined to yield enough nuclear protein for an immunoprecipitation and IgG sample. Pooling these lysates resulted in 1320ug of cytosolic protein in 1mL and 240ug of nuclear protein in 500uL. Taking 110ug of protein for each immunoprecipitation and IgG sample thus resulted in a significantly larger fraction (~50%) of nuclear lysates being used for immunoprecipitation compared to cytosolic lysates (~8%). This discrepancy in the fraction of lysate used to pull down TG2 likely explains the weak pull down of TG2 from the cytosol observed. To control for this, we ran another immunoprecipitation experiment using the exact same cytosolic lysate from the original pulldown. In the new pulldown 400ug of cytosolic protein was used (~30% of the original total) for each immunoprecipitation and IgG sample. We have adjusted Fig3d to include these more equitable cytosolic samples.

Minor comments, and questions which could contribute to the improvement of the manuscript:

  1. Please, make it more clear in the abstract (and in the title), that the study is based on mouse cells/model.

We have now clearly stated that we used mouse astrocytes in the title and abstract. In the abstract we describe the cell model.

  1. The number of independent repetitions of experiments is not obvious. Please, label in the text, or method or figure legends where it is applicable.

We have now included the number of independent biological experiments accounted for by the total number of technical replicates in each figure.

  1. The list of abbreviations or completion of figure legends with them would make easier the understanding.

We have now ensured all abbreviations used in each figure are explained in the associated figure legend.

  1. Labeling the molecular weight of protein markers is necessary to judge the original images due to probably unspecific bands.

We have now included the molecular weight standards on the uncropped blots.

  1. Some additional explanations and labels would help the complete understanding of Yeast Two-Hybrid results in the supplement.

Additional description of the Y2H data has been added to the supplemental figure.

  1. Page 8 lane 231-232: Maybe it could be written: ”…, but not so significantly,…”. based on the results in Fig5b (* vs. ***).

We have now reworded this statement.

  1. It is not completely clear, why has TG2 a necessarily inhibitory effect on gene expression. Is it possible, that the reason for increased gene expression is compensation due to the loss of TG2?

This is a possibility, however the data overall indicate that TG2 downregulates gene expression. For example, astrocytic TG2 depletion increases nuclear factor-κB (NF-κB) signaling, whereas enzymatic inhibition reduces NF-κB activity, suggesting that non-enzymatic TG2 interactions facilitate transcriptional inhibition  (Feola, et al., 2017). We have also examined the effect of TG2 using other promoters in the luciferase assay and find that TG2 can repress activity. Further, TG2 localizes to the nucleus and is found predominantly in the chromatin fraction, and nuclear localization of TG2 is required for its suppressive effect on HIF activity (Gundemir et al., 2013). These are just a few examples indicating that TG2 inhibits gene expression.

Reviewer 2 Report

Deletion of transglutaminase 2 from astrocytes significantly improves their ability to promote neurite outgrowth on an inhibitory matrix.

The authors in the current manuscript determine the role of Transglutaminase 2 (TG2) expressed in astrocytes in contributing to neuronal growth. The in vitro work presented in the current manuscript is based on their in -vivo observations where in TG2 deletion selectively in astrocytes improves functional recovery in ischemic brain and spinal cord injury models.  

The authors have used a trans- well, co-culture system involving, primary cortical neurons (E18; SD rats) and primary cortical astrocytes (WT or TG2-/- C57BL/6 mice). Neuron length, total neurites and ability to form synapses with astrocytes are functional endpoints are used to determine the neuronal growth. The neurons are subjected to growth in an ECM that is friendly (PDL) or an ECM that is inhibitory (CSPG).  Astrocytes cytoskeletal remodeling as measured by GFAP network is a measure for astrocyte reactivity. The authors propose that TG2 deletion in astrocytes improves neuronal growth only in 1) inhibiting matrix 2) when the neurons are seeded in lower densities in the inhibitory matrix. This function of TG2 is primarily attributed to its repressor function mediated by binding to transcription factor Zbtb7a in the nucleus and inhibiting the beneficial effects of Zbtb7a in neuronal growth.

Major comments:

The manuscript has been presented in a very disorganized way. The figure legends are either incomplete or inaccurate. Supplementary figures are provided but do not have details or figure legends. Thus, making it difficult to understand the scientific rigor applied for presenting this manuscript for a peer review.

§  Figure 1: It would help to present the data from WT and TGF2-/- only astrocytes on neuronal growth here. The labeling is very poor. What NA represents in not clear from the figure legend or the manuscript. How do the authors explain the observation that TG2 neurons show increase in total length however no change in total neurites? What was the statistics used for analysis?

§  Figure 2: A) neurons cultured in PDL matrix. B) neurons cultured in CSPG matrix: From the representative figures provided neurons grown on CSPG look healthier. Further neurons co-cultures in the presence of TG2-/- astrocytes appear stressed. What is the difference between the two bar graphs presented in C)? Labeling incomplete in IFC images presented in D). What was the rationale behind using PDL and CSPG matrix together? Please mention the statistical analysis performed.

§  Figure 3: labeling again in complete. A) Is this nucleus and cytosol fraction? Please use a better nuclear marker. There is no co-localization data in this figure. How was the GFAP network quantified?

§  Figure 4: Legend title and legend details do not match. Labeling incomplete.

§  Figure 5: It is interesting to notice that in Figure 3 Zbtb7a overexpression increases the GFAP network and then in Figure 5 Zbtb7a also increases neurite outgrowth. How do the authors explain this discrepancy? Have the authors observed the effect of Zbtb7a overexpression on neuronal growth in inhibitory matrix environment.

§  TG2 is also secreted. Have the authors tried out performing these experiments in the presence of exogenous TG2. This would be important for understanding if TG2 exerts in effects on neuron potentially via other mechanisms.

Author Response

Reviewer #2

The manuscript has been presented in a very disorganized way. The figure legends are either incomplete or inaccurate. Supplementary figures are provided but do not have details or figure legends. Thus, making it difficult to understand the scientific rigor applied for presenting this manuscript for a peer review.

The reviewer is correct that there were problems with the organization of the original submission and we apologize. We had difficult in correctly positioning the multipanel micrographs in the template. We have now spent a considerable amount of time making sure that the manuscript is correctly organized.

  • Figure 1: It would help to present the data from WT and TGF2-/- only astrocytes on neuronal growth here. The labeling is very poor. What NA represents in not clear from the figure legend or the manuscript. How do the authors explain the observation that TG2 neurons show increase in total length however no change in total neurites? What was the statistics used for analysis?

We have now defined all abbreviations in the legends and included information about the statistics used.

  • Figure 2: A) neurons cultured in PDL matrix. B) neurons cultured in CSPG matrix: From the representative figures provided neurons grown on CSPG look healthier. Further neurons co-cultures in the presence of TG2-/- astrocytes appear stressed. What is the difference between the two bar graphs presented in C)? Labeling incomplete in IFC images presented in D). What was the rationale behind using PDL and CSPG matrix together? Please mention the statistical analysis performed.

We have improved the presentation of Figure 2. In 2(c) the left graph is just of PSD95 spots co-located with Synaptophysin, the right graph is using intensity based identification of spots to more accurate capture puncta in the CSPG condition. This information is now in the legend.

  • Figure 3: labeling again in complete. A) Is this nucleus and cytosol fraction? Please use a better nuclear marker. There is no co-localization data in this figure. How was the GFAP network quantified?

We have revised Figure 3 and the legend. Information about quantification of the GFAP network is describe in the methods section.

  • Figure 4: Legend title and legend details do not match. Labeling incomplete.

The legend and legend details for Figure 4 are now complete.

  • Figure 5: It is interesting to notice that in Figure 3 Zbtb7a overexpression increases the GFAP network and then in Figure 5 Zbtb7a also increases neurite outgrowth. How do the authors explain this discrepancy? Have the authors observed the effect of Zbtb7a overexpression on neuronal growth in inhibitory matrix

It is unclear why the increase in GFAP network and the ability to promote neurite outgrowth due to overexpression of Zbtb7a in WT astrocytes would be considered discrepant results. At this time we have not carried out studies using neurons on an inhibitory matrix with astrocytes in which Zbtb7a expression has been manipulated.

  • TG2 is also secreted. Have the authors tried out performing these experiments in the presence of exogenous TG2. This would be important for understanding if TG2 exerts in effects on neuron potentially via other mechanisms.

It is true that astrocytes can secrete TG2 and in the future we certainly will consider using exogenous TG2 in the neurite outgrowth model.

Reviewer 3 Report

The manuscript submitted by Jacen Emerson and coauthors demonstrates the role of transglutaminase 2 in astrocyte functioning in health and disease. Great attention has been paid to such important function of the astrocytes as synaptogenesis regulation. Some key conclusions have been made based on the data obtained by immunostaining techniques. However, I have some concerns about specificity of the used antibodies or staining protocols.

 Major points

1. It is well known that Synaptophysin is a quite reliable presynaptic marker. Nevertheless, we can see bright, evenly distributed fluorescence in neuronal soma in addition to puncta on the neuronal processes. This fact indicate non-specific binding or inappropriate staining protocol. The similar conclusion can be drawn from the results of anti-PSD95 staining. Please, explain these artifacts. Positive and negative controls demonstrating specificity of the antibodies are required.

2. The morphology of the cell cultures also arises numerous questions. First of all, the authors demonstrate single neurons on the figures, but describe the used object as "cultures". Please, show bright-field (transmitted-light) images of the obtained cultures in order to estimate the culture density. Since synaptogenesis and neurite outgrowth in the culture are strongly depend on the microenvironment, cell density may significantly affect the results and their interpretation.

3. In addition to morphology of neurons, the attention also should be paid to astrocyte morphology. As known, the astrocytes in brain slices or co-cultures are "star-shaped" cells with numerous processes containing GFAP. As we can see in the figures, GFAP is demonstrated in the soma, while the processes almost absent. Please, clarify this point and demonstrate the astrocyte morphology using lower magnification.  

 Minor points

-Please, revise the abstract to make it shorten.

-Although the used statistical tests are described in Materials and methods section, it would be better including this information to the figure captions.

Author Response

Reviewer #3

  1. It is well known that Synaptophysin is a quite reliable presynaptic marker. Nevertheless, we can see bright, evenly distributed fluorescence in neuronal soma in addition to puncta on the neuronal processes. This fact indicate non-specific binding or inappropriate staining protocol. The similar conclusion can be drawn from the results of anti-PSD95 staining. Please, explain these artifacts. Positive and negative controls demonstrating specificity of the antibodies are required.

The fluorescence in the soma was initially a concern for us as well. However, upon close inspection, the fluorescence is not evenly distributed, but actually punctate. This is much easier to determine at lower levels of brightness and saturation. Additionally, this soma pattern is consistent across all synaptic markers used (including Bassoon and Homer). We also searched through the literature and found similar examples of soma fluorescence with PSD-95 and Synaptophysin (PMID: 12151521, PMID: 15358863). Based on this presentation, we assume that synaptic proteins are produced in the cell soma and trafficked down to its processes. This appearance may also be dependent on antibody specificity for synaptic marker isotypes. Since fluorescent spots cannot be accurately quantified from the soma, we masked and subtracted this signal and only quantified puncta in the processes. Negative controls (primary only and secondary only) were used for initial testing of antibodies, which showed no fluorescent signal. These can be repeated if needed.

  1. The morphology of the cell cultures also arises numerous questions. First of all, the authors demonstrate single neurons on the figures, but describe the used object as "cultures". Please, show bright-field (transmitted-light) images of the obtained cultures in order to estimate the culture density. Since synaptogenesis and neurite outgrowth in the culture are strongly depend on the microenvironment, cell density may significantly affect the results and their interpretation.

We have now added representative wide-field view images of the glass coverslips used for our “high seeding density” neurite outgrowth experiment as supplemental figures. Cell density certainly has a strong influence on the outputs of our experiments. We found this by repeating the experiment in Figure 1 with 2x the seeding density, which diminished any significant effects we found in the original design. Synapse studies used the same seeding density as this “high-density” neurite outgrowth experiment. However, we are able to isolate individual neurons for imaging as fixing and processing steps wash off cells to a greater degree for neuron cultures at 12 days in vitro (synapse experiments), with more extensive networks, compared to cultures at 5 days in vitro (neurite outgrowth), which tend to resist loss of cells through processing.

  1. In addition to morphology of neurons, the attention also should be paid to astrocyte morphology. As known, the astrocytes in brain slices or co-cultures are "star-shaped" cells with numerous processes containing GFAP. As we can see in the figures, GFAP is demonstrated in the soma, while the processes almost absent. Please, clarify this point and demonstrate the astrocyte morphology using lower magnification.  

In vitro, astrocytes are commonly cultured in serum-containing media(PMID: 6248568), which makes them take on a reactive phenotype that often looks polygonal or sometimes spindle-shaped in morphology. This is the common morphology we find in our cultures. Figure 3b is the most representative example. In Figure 4b, areas of dense GFAP accumulation at the cell membrane appear to be sites for focal adhesion complexes, which are also common. The star shape is classic for astrocytes in vivo and can be replicated by using a defined serum-free media in culture, supplemented with specific growth factors (PMID: 21903074). While the range of reactive states an astrocyte can take is not well understood, even in vivo, we believe this standard culture method works in our favor to upregulate TG2 in wild type astrocytes, further exaggerating the effect of TG2 in astrocytes. Models of CNS injury share this upregulation of TG2 in astrocytes (PMID: 9013629 , PMID: 20731658). We have compared TG2 expression in astrocytes cultured in these two media types, and validated an upregulation of TG2 in the standard, serum-containing media by qPCR.  

 Minor points

-Please, revise the abstract to make it shorten.

The abstract has been shortened.

-Although the used statistical tests are described in Materials and methods section, it would be better including this information to the figure captions.

We have now included information about the statistical tests in the legends.

Round 2

Reviewer 1 Report

In my opinion, the content of the manuscript is ready for publication.

If it is accepted, I suggest a meticulous check. In the provided pdf version very hard to see which part is really deleted and included, word doubling could occur.

Reviewer 3 Report

The authors have addressed all my comments.